# An Update on the Interplay between LRRK2, Rab GTPases and Parkinson’s Disease

**DOI:** 10.3390/biom13111645

**Published:** 2023-11-13

**Authors:** Tadayuki Komori, Tomoki Kuwahara

**Affiliations:** Department of Neuropathology, Graduate School of Medicine, The University of Tokyo, Tokyo 113-0033, Japan

**Keywords:** LRRK2, Rab, endolysosome, intracellular transport, Parkinson’s disease

## Abstract

Over the last decades, research on the pathobiology of neurodegenerative diseases has greatly evolved, revealing potential targets and mechanisms linked to their pathogenesis. Parkinson’s disease (PD) is no exception, and recent studies point to the involvement of endolysosomal defects in PD. The endolysosomal system, which tightly controls a flow of endocytosed vesicles targeted either for degradation or recycling, is regulated by a number of Rab GTPases. Their associations with leucine-rich repeat kinase 2 (LRRK2), a major causative and risk protein of PD, has also been one of the hot topics in the field. Understanding their interactions and functions is critical for unraveling their contribution to PD pathogenesis. In this review, we summarize recent studies on LRRK2 and Rab GTPases and attempt to provide more insight into the interaction of LRRK2 with each Rab and its relationship to PD.

## 1. Introduction

Since its discovery as the protein responsible for Parkinson’s disease (PD) in the *PARK8* locus in 2004 [1,2], leucine-rich repeat kinase 2 (LRRK2) has been one of the main focus molecules associated with this neurodegenerative disease. The physiological roles of LRRK2 have been linked to a myriad of cellular processes, such as several types of autophagy including macroautophagy and mitophagy [3,4], endocytosis and intracellular transport involving the *trans*-Golgi network (TGN) and other organelles [5,6,7,8], the regulation of microtubules [5,6], interaction with bacterial pathogens [7], regulation of lysosomal homeostasis [8], and much more.

The pathogenic features of LRRK2 have also been studied by analyzing mutations associated with PD, and most, if not all, mutations point towards a similar effect: augmentation of substrate phosphorylation [8,9]. Albeit these findings, the exact mechanism of how defects in LRRK2 lead to PD has been elusive for nearly a score of years now. An auspicious approach to this enigma would be its link to the endolysosomal system and their regulators, Rab GTPases, as a considerable number of findings indicate connections between these [10]. 

Rab GTPases bind to membranes and utilize their affinity change via their guanine nucleotide binding status to form specific functional domains on their corresponding organelle membranes [11], hence called the master regulators of intracellular vesicular traffic. Some of the first reports that related Rab GTPases to PD were around 20 years ago, when α-synuclein was reported to interact with several Rab GTPases [12] or α-synuclein-induced neuronal loss was rescued by overexpression of Rab1 [13]. Research was boosted when LRRK2 was found to be associated with another candidate PD risk gene, Rab29 (also known as Rab7L1) [10], and by subsequent findings of Rab phosphorylation by LRRK2 in cells [14,15,16,17,18,19]. Recent advances in structural analysis have also revealed several interesting findings and insights about LRRK2 and its functions, with some of them incorporating Rab GTPases as its interactor. In this review, we aim to provide an update of the current findings on LRRK2, both from structural and functional aspects, as well as known interacting Rab GTPases and their functions, hoping to give a better view on LRRK2, Rab GTPases, and PD. 

## 2. Insights from Genetic and Structural Studies of LRRK2

Back in 2002, Funayama et al. reported a family of inherited PD with an unknown causative gene locus, *PARK8* [20]. Two years later, two independent groups identified the responsible gene as *LRRK2* [1,2]. Since then, genetics have revealed at least seven causative mutations in this gene product: N1437H [21], R1441C [1,22], R1441G [22], R1441H [22], Y1699C [1], G2019S [23], I2020T [1], and numerous other rare variants with unclear causality [24]. The gene product LRRK2 is a 2527-amino-acid multidomain kinase, and the PD-associated mutations lie in the ROC (Ras of complex), COR (C-terminal of ROC), and kinase domains (Figure 1). The ROC domain is a GTP-binding domain that regulates the kinase activity in an intramolecular fashion [25,26], although its GTP-binding state (not GTP-binding capacity) may not be required for its kinase activity [27]. The ROC domain also works as a scaffold for protein interactions [28], whereas the COR domain constitutes an interface for LRRK2 dimerization [29,30]. All the aforementioned LRRK2 mutations result in an increase in phosphorylated substrates [18,31,32], which is not necessarily accompanied by an increased kinase activity of LRRK2 itself [29,30].

Other domains in LRRK2 have no PD-associated mutants allocated to them but are also of importance when considering intermolecular interactions. Armadillo repeats (ARM), ankyrin repeats (ANK), and leucine-rich repeats (LRR) are relatively abundant motifs that form various sizes of scaffolds for protein interaction [33,34,35], whereas WD40 domains are beta-propellers that may interact with DNA as well as proteins [36,37]. The detailed primary structure is depicted in Figure 1A. It might also be worth noting that a portion of endogenous LRRK2 in macrophages is found cleaved at the ANK-LRR interdomain region to produce a C-terminal fragment including the kinase region [38]. Although this fragment may be nonfunctional because the N-terminal membrane-interacting region is lacking, it may also act as dominant negative as it can heterodimerize with full-length LRRK2 [38].

**Figure 1 biomolecules-13-01645-f001:**
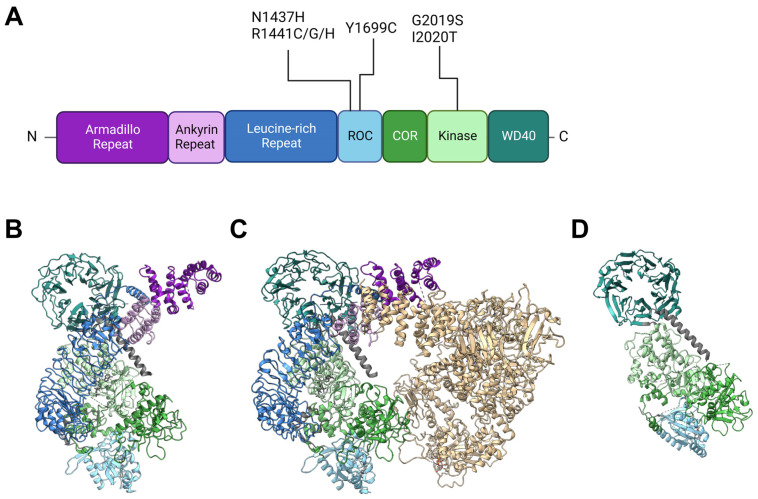
Structures of LRRK2 and PD-associated mutations. (**A**) Primary structure of LRRK2 and the location of reported PD-associated mutations. ROC: Ras of complex domain, COR: C terminal of ROC domain, WD40: WD40 domain. Domain lengths and borders are based on [39]. (**B**) Monomeric structure of full-length LRRK2 [30] (PDB ID: 7HLT). (**C**) Homodimeric structure of full-length LRRK2 [30] (PDB ID: 7HLW). (**D**) Protomer of filamentous LRRK2 C-terminal half (RCKW) on microtubules [40] (PDB ID: 6VNO). All structures are aligned so that the WD40 domain lies at the top left. Colorization of structures is the same for (**A**–**D**).

LRRK2 is observed in many conformations in vitro or in cells, with structural models from monomer [30] to homodimer [30,41,42] to filamentous [40,43] and even heteromultimer [44] reported (Table 1, Figure 1B–D). The classical model of LRRK2 is the homodimer model, based on biochemical analysis and crystal structures of ROC domains [45,46]. This model was further confirmed via other methods such as cross-linking and negative stain [47], and further with full-length LRRK2 [30] (Figure 1B). 

The monomeric model was first proposed based on biochemical analysis of full-length LRRK2 [50] but was not confirmed until some years ago when the structure of full-length LRRK2 was finally determined via cryoelectron microscopy (cryo-EM) [30] (Figure 1C).

The filamentous LRRK2 model is based on observations of overexpressed PD-associated mutants, which are mostly cytosolic but often form filaments along microtubules [51]. Reports that analyzed this form of LRRK2 bound to microtubules via cryo-EM or cryoelectron tomography (cryo-ET) showed multimer formation via interactions through the WD40 domain with the help of microtubule filaments [40,43] (a protomer of the filament is shown in Figure 1D).

The analyzed structures of monomer or dimer forms of LRRK2 via Cryo-EM were reported to be captured in a kinase-inactive state [30,40], whereas another study utilizing microscale thermophoresis-based biophysical methods reported one of the active forms that bind either Rab8 or Rab10 [52]. Another cryo-EM analysis also observed a LRRK2-Rab29 complex and a tetrameric assembly of this complex, yielding active LRRK2 [44]. Of note, more recent studies have identified Rab12 rather than Rab29 as an activator of LRRK2, as described later, so a heteromultimer complex with Rab12 would also need to be assumed [53,54]. From these reports, one can infer that there are multiple modes of activation, and the activity should be broken down into several aspects. Indeed, we reported that the increased phosphorylation of Rab GTPases by LRRK2 under lysosomal stress can be separated into at least two phases: intrinsic LRRK2 activation and other means of accelerated interaction between LRRK2 and substrate Rab [55].

Interestingly, it has been shown that when LRRK2 is in the monomeric inactive state, interactions between the WD40 domains of LRRK2 and other molecules are blocked by parts of the LRR and ARM domains [30]. Other studies focused on how proteins could interact with LRRK2 via the WD40 domain of LRRK2, such as Syntaxin1 and NSF in synaptic vesicles [56], and these could also be tightly regulated by conformational changes by either protein modifications or proteins binding to LRRK2 in other domains. 

Either way, the regulation of LRRK2 activation relies partly on Rab GTPases, and revealing their cooperative mode of action is critical for unraveling the physiological and pathological roles of LRRK2.

## 3. Rab GTPases and LRRK2

### 3.1. Rab GTPases and the Endolysosomal System

Rab GTPases are proteins that form a subfamily of the Ras superfamily and are capable of binding cellular membranes via the C-terminal prenylation. Like other small GTPases, they switch their activity, which influences their affinity to specific proteins called effectors by changing their binding state with either GTP (active) or GDP (inactive) with the help of their specific guanine nucleotide exchange factors (GEF) or GTPase-activating protein (GAP). Their functions are associated with a wide variety of intracellular trafficking, ranging from cellular secretory pathways to intracellular degradation pathways involving the endolysosomal system.

The endolysosomal system is part of an intracellular flow of enveloped membrane organelles and, apart from its function in autophagy, serves as a sorting site for substances incorporated by endocytosis. These substances include extracellular materials as well as membrane proteins and cellular membranes themselves. These substances are then either guided towards degradation by lysosomes, returned (or “recycled”) to the plasma membrane, or routed to the TGN. This pathway is known to regulate basic steps of cellular processes such as signaling, adhesion, immunity, nutrient uptake, organelle homeostasis, membrane protein turnover, and much more (reviewed in [57,58,59,60,61,62]).

This route to degradation can be broken up into several parts, with more than one Rab GTPase regulating the routing or maturation of each vesicle. Endocytosed materials are first retained in the early endosome, where Rab5 controls the maturation of the vesicle and other Rab GTPases, such as Rab11, regulate re-routing from the early endosome to other compartments, in this case to the recycling endosomes, another part of the endolysosomal system, and ultimately to the plasma membrane [63,64]. Membranous proteins targeted for degradation start to form vesicles called intraluminal vesicles (ILVs) inside of this early endosome with the help of some specialized protein complexes [57,61]. The formation of ILVs continues throughout the maturation process. Matured early endosomes transform into slightly more acidic vesicles with different lipid compositions, which are called late endosomes. At the late endosomes, Rab7 is the controller of further maturation, and again other Rab GTPases, such as Rab9, account for the re-routing to other compartments [63,65]. Eventually, the late endosomes, with all the reusable proteins routed away to the plasma membrane or *trans*-Golgi network, gain the ability to fuse with lysosomes with the help of several Rab7 effector proteins, marking the last part of the endolysosomal system [60,63]. In some cells, Rab GTPases also regulate cell-type specific vesicles such as synaptic vesicles in neurons and melanosomes in melanocytes [66,67].

### 3.2. LRRK2 and Substrate Rab GTPases

As a kinase, LRRK2 phosphorylates a subset of Rab GTPases, e.g., Rab8 and Rab10, in cells. In this section, we review the functions and pathobiology of each Rab GTPase, citing some new reports that may clarify their relationship to LRRK2 and their contribution to PD. The overview of the Rabs with their roles upon phosphorylation is summarized in Figure 2 and Table 2.

#### 3.2.1. Rab8 and Rab10

Rab8 and Rab10 are closely related Rabs, both categorized in the Rab8 subfamily [89] and are the most characterized Rab GTPase in the context of interaction with LRRK2. 

Right after the initial report of multiple Rab phosphorylation by LRRK2 [14], Rab10 was found to be a very sensitive marker for assessing LRRK2 activity [90], followed by a quick development of a phospho-specific antibody against Rab10 [68]. To date, numerous studies have incorporated an assessment of this phosphorylation in their studies on LRRK2 kinase activity [24,55,76,77,78,81,82,85,91,92,93,94,95,96,97,98,99]. Moreover, several reports have attempted to utilize phosphorylated Rab10 as a biomarker for upregulated LRRK2 activities in PD or preclinical models, which was shown to be successful to a certain extent, although results vary between studies [100,101,102,103,104,105,106,107,108,109]. Although it is almost established that the phosphorylation state of Rab10 reflects the activity of LRRK2, there are still some function-related ambiguities of this GTPase that have to be straightened out.

Rab10 has been implicated in several modes of transport in a variety of cell types, from general exocytic pathways to neurite or cilia formation and immune responses [110]. LRRK2 has been shown to play a role in these functions via its kinase activity, altering the ability of Rab10 to interact with other proteins. In the context of ciliogenesis, phosphorylated Rab10 binds to RILPL1 at or near centrosomes, inhibiting ciliogenesis [15,71,78,79,80]. This inhibition was also brought about by the retention of Myosin Va at centrioles by the same phosphorylation of Rab10 [77]. In the context of endosomal trafficking, phosphorylation of Rab10 at steady state on early macropinosomal membranes results in a decrease in binding to EHBP1L1, a Rab10 effector. This then inhibits the recycling of macropinosomes, ultimately hindering chemotaxis in macrophage cells [81]. Another effect on endosomal trafficking is in lysosomal stress-induced cells, where Rab10 as well as its effectors EHBP1 and EHBP1L1 regulate lysosomal content release [76]. Other functions include lysosomal tubular budding named LYTL (LYsosomal Tubulation/sorting driven by LRRK2) upon Rab10 phosphorylation by LRRK2 and recruitment of JIP4 [82,83], negative regulation of a lysosomal enzyme CGase upon Rab10 phosphorylation by LRRK2 [84], and induction of lysosomal overload stress and apoptosis after neuronal injury by the same phosphorylation [85].

Rab8, on the other hand, is well characterized as a Rab GTPase, with GEFs Rabin8 and GRAB, GAPs TBC1D30, and other TBC family proteins, and multiple effectors described in the context of anterograde trafficking, endocytic recycling and exocytosis, association with the cytoskeleton, cell shape regulation and migration, ciliogenesis, neurite growth, and much more [111]. Although there have been attempts to utilize the phosphorylation of this GTPase as a biomarker for PD [112], further studies are still required for the assessment of Rab8 phosphorylation, as the currently available antibody against Rab8 phosphorylated at Thr72 (MJF-R20) has been shown to cross-react with phosphorylated Rab3A, Rab10, Rab35, and Rab43 [68].

The first reports on Rab8 functions date back about 30 years when Rab8 was deemed responsible for post-Golgi anterograde trafficking in epithelial and neuronal cells [113,114]. Studies on Rab8’s involvement in neurite formation immediately followed [115] and built the classical view of Rab8 as a controller of neurite formation via anterograde trafficking. Current knowledge on Rab8 in neurite formation includes additional upstream elements, involving various other Rabs and their effectors, such as Rab11 and Rabin8, that activate Rab8 and Rab10 for neurite outgrowth [116], as well as downstream elements such as Cdc42 and tuba that strictly regulate the number of axons formed per cell [117]. 

Another aspect of Rab8 is its involvement in cilia. Formation of cilia requires Rab8 activation on the centrosome by Rabin8, very much like in neurite formation, and further protein trafficking to the base of primary cilia [111]. Impaired receptor trafficking to cilia via Rab8 dysfunction causes various deficits in functions that require cilia, such as adipocyte differentiation, where Rab8 is responsible for the trafficking of frz2 to the base of primary cilia [118]. Rab10 and Rab13 might have compensatory roles in ciliogenesis as double knockouts of Rab8a and Rab8b were insufficient to cause cilial deficits [119].

These functions of Rab8 rely on proper trafficking from the TGN or recycling endosomes to each responsible compartment, most likely being the central function of Rab8 shared among various cell types.

Some reports have also attempted to delineate the relationship between Rab8 and PD by hypothesizing a direct interaction between Rab8 and α-synuclein, a protein well known to both cause PD and accumulate in PD brains [12,120]. They seem to bind to each other in vitro [12] at the switch region of Rab8 [120], and their binding enhances α-synuclein fibrilization while ameliorating its toxicity [120]. Although the detailed mechanisms are yet to be unraveled, a model that suggests that Rab8, together with optineurin, a Rab8 effector, is involved in the initiation of autophagy near aggregated proteins [121] might also give us insight. Altogether, it is possible that Rab8 is related to the pathogenesis of PD in some way.

Although most of the above-mentioned functions of Rab8 and Rab10 indicate a very close and very much overlapping role for the two Rabs, there are reports that suggest an antagonistic property for them. One report depicts a discrepancy in the role behind ciliogenesis, where overexpression of GFP-tagged Rab8 exhibited an increase in cilia while overexpression of GFP-tagged Rab10 had the opposite effect [79]. Another recent report has also examined the differential functions of Rab8 and Rab10. Although knockouts of either Rab8 or Rab10 show defects in lysosomal or Golgi homeostasis, such as a decrease in lysosomal number or dispersion of the Golgi, a further look into the details revealed some divergent phenotypes: Rab8 knockouts had acidic lysosomes whereas Rab10 knockouts had elevated pH [122]. These reports collectively suggest that Rab8 and Rab10 control Golgi- and endosome-related trafficking from slightly different aspects, some resulting in seemingly opposite phenotypes when suppressed or overexpressed.

Things get more complicated when their kinase LRRK2 cuts in. Some reports suggest that the phosphorylation of Rab8 inhibits its function and causes centrosomal or ciliogenesis deficits [17,71,72] or decreased binding with its effector [73], while other reports indicate new functions such as the recruitment of RILPL2 [74] or alteration of the endosomal pathway [75]. Either way, phosphorylation of Rab8 by LRRK2 seems to alter its binding ability to effectors, just like in Rab10, leading to the observed changes in its function.

We have previously reported a difference in the functions of Rab8 and Rab10 downstream of LRRK2 [76]. Lysosomal enlargement is an outcome of mild lysosomal stress that is not enough to result in lysosomal rupture, and the resultant cellular responses include the deflation of enlarged lysosomes and the promotion of the extracellular release of lysosomal contents. Depletion of Rab8 enhanced lysosomal enlargement, whereas that of Rab10 diminished the exocytosis of lysosomal contents [76]. These lysosomal changes were further controlled by their effectors EHBP1 and EHBP1L1, effectors of Rab10 and Rab8a, as depletion of these effector proteins resulted in both enhanced lysosomal enlargement and lowered lysosomal exocytosis [76]. This may suggest that Rab functions could be defined not only by a single effector but also by the combination of multiple proteins. Indeed, EHBP1 has been reported to bind to both Rab10 and EHD2 to form a ternary complex [123] or to bind active Rab8 and be activated as a scaffold to bind additional proteins such as f-actin [124]. These reports indicate a more complex mode of action: a Rab-and-effector-based recognition of, or an affinity change to, a third factor. Investigating changes in the binding ability of EHBP1/EHBP1L1 upon their binding to Rab8/10 could be worthwhile to further delineate the intertwined functions of the two Rabs.

#### 3.2.2. Rab29

Also known as Rab7L1, Rab29 itself is nominated as a risk factor for PD, encoded in the *PARK16* locus [125,126,127]. This GTPase was among the first of all Rabs to have their relationship with LRRK2 uncovered [128,129] and is the Rab that is known to regulate LRRK2 from upstream [16,72,76,87,92,130] in addition to the recently reported Rab12 [53,54] and Rab38 [131]. Rab29 localization at steady state was reported to be at the Golgi, with a small fraction at perinuclear vesicles [132,133]. These Rab29 population at the Golgi was reported to be responsible for the integrity of the TGN and the retrograde trafficking there [18,134]. Golgi fragmentation was also dependent on the recruitment and activation of LRRK2 induced by Rab29 overexpression [16,19], as well as phosphorylation of Rab29 by LRRK2 [18].

The function of Rab29 has attracted attention not only from its relationship to the Golgi but also to the lysosome, as the small population of Rab29 at perinuclear vesicles was found to be lysosomal. Also, Rab29 has been shown to react to lysosomal stress, localizing itself to lysosomes and also co-recruiting and activating LRRK2 [76,87]. Active Rab29 on lysosomes regulates the size of abnormally inflated lysosomes, which is dependent on LRRK2 kinase activity [76,87]. The localization of Rab29 itself to lysosomes also depends on the activity of LRRK2 as well as some PKC isoforms and their ability to phosphorylate Rab29 at corresponding sites [87].

The physiological importance of the function of Rab29 on lysosomes and its relationship with LRRK2, at least in some respects, is confirmed by loss-of-function studies. Knockdown of endogenous Rab29 in macrophage cells causes excessive lysosomal enlargement and inhibits LRRK2 recruitment [76] and activation [55] upon lysosomal stress loading. Consistent with this, Rab29 knockout mice have been shown to exhibit the accumulation of enlarged lysosomes in renal proximal cells, which is strikingly similar to the renal phenotype reported in LRRK2 knockout animals [130]. On the other hand, several other studies have reported few or no phenotypes related to the regulation of LRRK2 activation in Rab29 knockout cells, including MEFs, lung cells [133], and HEK293FT cells [99]. Further studies would be needed to determine whether these differences in results are due to differences in cell types (macrophages vs. other cell types), different experimental conditions (e.g., transient knockdown vs. stable knockout), or other reasons. Considering the observations that LRRK2 is activated once recruited to membranes [92] and that LRRK2 harbors a site that strongly binds phosphorylated Rab substrates other than the site that binds Rab29 [52], Rab29 might be acting as an initiator of LRRK2 activation and not necessarily keeping LRRK2 active.

In any case, Rab29 behaves poorly at steady state [135] but is very reactive to lysosomal stress [76,87], which suggests a role in lysosomal troubleshooting. Although the exact stimulus or conditions that Rab29 reacts to, or their sensor proteins, still need closer investigation, Rab29 is surely a promising key molecule in deciphering lysosomal stress responses, if not PD pathogenesis.

#### 3.2.3. Rab12

Rab12 was first found to regulate a “non-canonical” degradation route from recycling endosomes directly to lysosomes [136], then further allocated to more transfer between the cell surface and Golgi for various cargoes [137,138]. Rab12 is also gradually being understood as a potent marker of LRRK2 activity, as the phosphorylation of Rab12 was reported to be potently induced by PD-associated mutants of LRRK2 [133,139]. The functions of the phosphorylation of this GTPase were not known until very recently; it was found to be responsible for controlling the intracellular localization of lysosomes via an increase in the binding ability to RILPL1 [86]. An unusual point about this phosphorylation is that LRRK2 recognizes GDP-bound Rab12 better than the GTP-bound form, at least in vitro [140]. Rab12 is also implicated in lysosomal repair, as Rab12 accumulates on damaged lysosomes and activates LRRK2 there [53,54]. Rab12-mediated accumulation and activation of LRRK2 on lysosomes during lysosomal damage were enhanced in PD-associated mutants of LRRK2 or even VPS35, even under non-damaged conditions, but were not enhanced beyond wild-type during damage [54], suggestive of a Rab12-dependent lysosomal response mechanism that might be constantly activated in the course of PD pathogenesis with these mutations. This accumulation and activation were brought about by Rab12 binding to LRRK2, and although the binding site on LRRK2 was located in its N-terminal Armadillo domain that includes the binding site of Rab29 or phosphorylated Rab8 or Rab10, the detailed site was different from these molecules [53]. This is indicative of yet another pathway in sensing lysosomal abnormalities apart from Rab29, and as they both augment LRRK2-induced phosphorylations, Rab12 would be a promising GTPase to dig into in terms of the pathogenesis of PD.

#### 3.2.4. Rab35

Rab35 is a Rab GTPase responsible for various cellular processes including exosome release, neurite outgrowth, phagocytosis, cell polarization, immune synapse formation, cytokinesis, and cell migration [141]. These pathways are controlled by either the quick recycling of endocytic cargoes (e.g., T cell receptor (TCR) and MHC complexes for immune synapse formation, podocalyxin for cell polarization) to the plasma membrane or the regulation of actin beneath the plasma membrane to promote changes in cell shape and position [141]. Although the main link to diseases would be between cancer, there are several reports that link this GTPase and LRRK2 to PD.

The effectors of Rab35 include OCRL, MICAL1, and MICAL-L1 [132], which are also effectors of Rab8 or Rab10 as noted above. Not surprisingly, some of the functions of Rab35 overlap with Rab8 and Rab10, which include the recruitment of JIP4 and induction of LYTL upon phosphorylation by LRRK2 [83]. LRRK2-induced phosphorylation of Rab35 was also found to positively regulate the propagation of α-synuclein [88]. This may be caused by either tubulation of lysosomes (LYTL) or the fast recycling of endocytic content, but details need further assessment. Nevertheless, Rab35 may be an important Rab in the development of PD, and possibly, in more initial pathogenic mechanisms.

#### 3.2.5. Rab5

Rab5 is the key GTPase in controlling the maturation of early endosomes. After endocytosis, Rab5 is recruited to the endocytic vesicle by Rab4 or the Rab5 GEF Rabex5 (RABGEF1), and then in turn recruits effector proteins such as EEA1, a tether for fusion with other early endosomes, VPS34, a phosphatidylinositol kinase responsible for converting phosphatidylinositol (PI) to the endosome-enriched lipid phosphatidylinositol-3-phosphate (PI3P), and the Rab7-GEF Mon1-Ccz1, which recruits Rab7 and facilitates transition to late endosomes [60].

Although there are many studies on Rab5, the differences between the three isoforms Rab5a, Rab5b, and Rab5c are not that undeciphered. Rab5c is reported to have a slightly different function from the other two, with little involvement in EGFR recycling [142] or specific involvement in Rac1-dependent cell migration [143]. Some more findings exist in non-mammalian cells, as Rab5a and Rab5b in *Leishmania* interact with different modes of endocytosis [135], or in yeast, ypt53 (one of the three Rab5 isoforms in yeast) is selectively upregulated under stressed conditions [144]. Further studies are expected.

Rab5 and PD have little connection reported, with implications in Rab5a-mediated uptake of α-synuclein in neurons [145] or clearance in microglia [146]. The latter is the phenotype also seen in LRRK2 knockout mice, which could hint at the possibility of Rab5 interplay in PD pathogenesis or treatment. Other links reported between LRRK2 and Rab5 include a cooperative regulation of neurite outgrowth [147], phosphorylation of all the isoforms of Rab5 by LRRK2 [15], and inactivation of Rab5b [70]. Note that phosphorylation of Rab5 isoforms by LRRK2 has only been observed upon overexpression in cells [15,68] or in vitro [70], and therefore we need to be cautious as to whether they are indeed physiological substrates. Nonetheless, these data could collectively outline a pathway from overactivated pathogenic LRRK2 to irregular inactivation of Rab5 and endosomal deficits, and although further confirmation is needed, Rab5 may be one of the key players behind PD development and progression.

#### 3.2.6. Rab3

Rab3 has four isoforms (Rab3a, Rab3b, Rab3c, Rab3d), and all of the isoforms participate in exocytosis or secretion. They are highly expressed in neurons and secretory cells [66,148]. Their roles in secretion in secretory cells or neurons appear to be redundant, with several knockout studies in mice observing little or no changes in exocytotic activity, but depletion of all Rab3 isoforms results in lethality from respiratory failure [149,150,151,152]. In neurons, Rab3 regulates a specific type of exocytotic vesicles called dense-core vesicles, which are important in neuropeptide release [153]. The four isoforms display different magnitudes of activity, with Rab3a being the most active [153]. Rab3a is also found to be responsible for plasma membrane repair via lysosomal exocytosis [154].

In cancer, these Rabs are associated with increased proliferation and invasion, possibly due to abnormal exosome release [155]. Here, Rab3d is the most popular isoform known to be upregulated in several types of tumors. Although research on Rab3 in non-secretory cells is limited, these perspectives might serve as hints to the functions of Rab3 in non-secretory cells.

There are several reports that associate Rab3a with α-synuclein [156,157,158]. Regulation of synaptic vesicle endocytosis is suggested to be a physiological function of α-synuclein, and disruption of this could very well be a cause of α-synuclein-related neurodegeneration [157].

Although Rab3 was identified as a LRRK2 substrate, there is only a limited number of studies on the interaction between these two proteins. One shows that Rab3a colocalizes with LRRK2 on stressed lysosomes dependent on its kinase activity [76], and another shows that Rab3 is a very weak substrate in LRRK2-G2019S expressing neurons, probably because of different localizations in neurons [159]. There is still quite a large gap between LRRK2, Rab3, and PD, but given the fact that Rab3 is implicated in lysosomal repair and interactions with α-synuclein, Rab3 could possibly be a potential factor in PD pathogenesis.

#### 3.2.7. Rab1

Rab1 is the newest Rab GTPase found to be a substrate of LRRK2 [69]. The classical role of Rab1 is its involvement in ER-Golgi transport and maintenance of the Golgi, but its functions reach out to regulating the localization of endosomes and lysosomes, and consequential cell-surface receptor recycling [160]. Loss of Rab1 results in fragmentation of the Golgi, which is seen in α-synuclein overexpression models [13] or in dopaminergic neurons in the substantia nigra of PD patients [161]. Another aspect of Rab1 is its involvement in autophagy, a cellular process in which cytosolic contents or specific organelles are sequestered in double-membrane autophagosomes and routed toward lysosomal degradation. Rab1 is necessary for the earliest steps in autophagy, possibly through the correct localization of autophagy-initiating proteins such as ATG9 [162,163,164]. Complete knockout of this Rab seems to be lethal [165], possibly due to its wide variety of functions it is responsible for.

It is yet to be determined how the LRRK2-induced phosphorylation affects Rab1 and its functions, but its involvement in PD pathogenesis would certainly be worthwhile to discover.

## 4. Rab Phosphorylation in Relation to PD

As stated above, almost all familial PD mutations increase Rab phosphorylation in cells, although the most common G2019S mutation may have somewhat different effects than others. That is, the G2019S mutation increases its intrinsic kinase activity more effectively than Rab phosphorylation, whereas the other mutations increase Rab phosphorylation more potently than its kinase activity [24]. This is consistent with the observations in human samples; the increase in Rab10 phosphorylation has been shown in peripheral blood neutrophils of R1441G mutation carriers [106], whereas G2019S mutation appears to have a weaker effect on Rab10 phosphorylation induction, at least in neutrophils [106] and peripheral blood mononuclear cells (PBMCs) [105,108]. These observations implicate slightly different pathomechanisms of PD for G2019S and other familial mutations.

In relevance to the pathomechanism of PD, one should take into account specific cell types in the brain, such as neurons and glia, as LRRK2 and each Rab are known to be expressed relatively ubiquitously in these cells. In analyses using mouse primary neurons and glia, Rab10 phosphorylation is detected in all cell types but is more strongly detected in astrocytes and microglia [105]. In vivo, it has been shown that cholinergic neurons in the striatum of LRRK2 R1441C knock-in mice develop ciliation defects, likely due to Rab10 over-phosphorylation [79], and a similar ciliation phenotype was subsequently observed also in astrocytes of LRRK2 G2019S knock-in mice [166]. Regarding the effects of Rab phosphorylation in microglia, it has been shown that microglial LRRK2 and Rab10 mediate manganese-induced inflammation and neurotoxicity [167]. In addition, since much of the work on LRRK2 and Rab has been conducted using macrophage cells like RAW264.7 cells, these findings would also apply to microglia, which are macrophage-lineage cells.

The actual changes of Rab10 phosphorylation in idiopathic PD (iPD) are also of interest but remain somewhat controversial. It has been shown that Rab10 phosphorylation is elevated in the substantia nigra [168] or PBMCs [108] of iPD patients as compared with healthy subjects. However, other groups have reported that the levels of Rab10 phosphorylation were similar between iPD and controls from the analysis of patients’ cingulate cortex [68] or in PBMCs/neutrophils [100]. Analysis of human urine has shown that urinary Rab10 phosphorylation is just barely higher in iPD than in controls (*p* = 0.046) [109]. Since the same group reported that the LRRK2 kinase activity itself in urine, as measured by the level of autophosphorylation at Ser1292, increased more clearly in iPD group (*p* < 0.01) [169], LRRK2 activation state and changes in Rab phosphorylation may not always coincide in PD.

There are some limitations to the above-mentioned analyses, and one should note that the changes of LRRK2 and Rab phosphorylation in PD are not yet at a stage to be fully discussed. First, the well-examined human biofluid samples to date are peripheral blood mononuclear cells and neutrophils, and these peripheral immune cells may exhibit different changes from those in the brain. Second, Rab phosphorylation status may vary greatly among individuals, and further classification of disease based on the type of progression or pathology, as well as other factors, will be necessary. Third, changes in Rab phosphorylation other than Rab10 should also be examined more systematically, but not much has been achieved yet as the generation of phospho-specific antibodies has lagged behind that of phospho-Rab10 antibodies. Fourth, immunoblot analysis after SDS-PAGE is often used for the assessment of Rab phosphorylation, but it would be desirable to develop a simpler and more sensitive assay for Rab phosphorylation detection. Lastly, in studies of PD, it will be required to determine at what time point around the onset of disease Rab phosphorylation is altered, and in which cell types this occurs specifically. Conducting large-scale studies recruiting a large number of PD patient samples, both idiopathic and familial, would address these points and better clarify the contribution of Rab phosphorylation to PD.

## 5. Conclusions and Future Perspectives

Rab GTPases have been established as the most promising and reliable substrates for LRRK2 since 2016, which has led to significant progress in LRRK2 research. In particular, recent structural and biochemical analyses are revealing more detailed modes of interaction between LRRK2 and Rab, which would advance our understanding of the regulatory mechanisms of Rab phosphorylation.

From a cell biology viewpoint, it would be interesting to further clarify what Rab over-phosphorylation causes in cells. As only a small fraction of Rabs is phosphorylated at steady state, it is likely that the majority of a given pool of Rab proteins can still carry out their normal functions [90]. On the other hand, phosphorylated Rab8a and Rab10 have been shown to bind new effector proteins, i.e., RILPL1/2 and JIP3/4 [15,74,83], suggesting that even a small fraction of them may dominantly affect cellular functions. Indeed, it has been shown that the recruitment of RILPL1 regulates ciliogenesis and centrosomal cohesion [15,71] and that of JIP4 regulates lysosomal tubulation [83] and axonal autophagosome transport [170]. In addition, as noted above, Rab10 phosphorylation by LRRK2 is markedly enhanced under lysosomal stress and is therefore assumed to play important roles in the maintenance of endolysosomes. The effects on disease-causing aggregate-prone proteins are also of interest; the brains of patients with LRRK2 mutations often, but not always, accumulate insoluble α-synuclein and tau to varying degrees, leading to the notion that Rab over-phosphorylation may also affect the metabolism or propagation of these proteins. Indeed, as mentioned above, phosphorylation of Rab35 has been shown to potentially regulate α-synuclein propagation [88]. However, the possibility remains that other Rabs may also be involved in different ways, and it is still unclear how and at what stage Rab over-phosphorylation influences the cellular dynamics of highly insoluble proteins.

With respect to clinical applications, LRRK2 inhibitors including small molecule compounds and antisense oligonucleotides are being developed, and one of them, BIIB122/DNL151, originally developed by Denali Therapeutics Inc, has now proceeded to phase III trials [171]. According to their clinical trial reports, administration of this LRRK2 inhibitor has been shown to markedly reduce Rab10 phosphorylation in PBMCs, as well as phospho-Ser935 LRRK2 in whole blood, total LRRK2 in cerebrospinal fluid (CSF), and a lysosomal lipid di-22:6-bis (monoacylglycerol) phosphate (BMP) in urine, all in a dose-dependent manner [172]. Also, the clinical trial results of another LRRK2 inhibitor, DNL201, were reported earlier, and the results were mostly similar [107]. In addition to LRRK2 inhibitors, some kind of Rab modulators may also be useful if a Rab acting downstream is identified as particularly important. Furthermore, targeting of LRRK2 and Rab phosphorylation in clinical applications may not be limited to PD, since LRRK2 is well-known as a risk gene for several immune/inflammatory diseases, including Crohn’s disease and leprosy. Given that LRRK2 is highly expressed in immune cells, cell type-specific effects of Rab over-phosphorylation would need to be clarified. The full picture of Rab phosphorylation by LRRK2 and its regulatory mechanism will hopefully lead to further elucidation of the pathomechanism of these diseases.

## Figures and Tables

**Figure 2 biomolecules-13-01645-f002:**
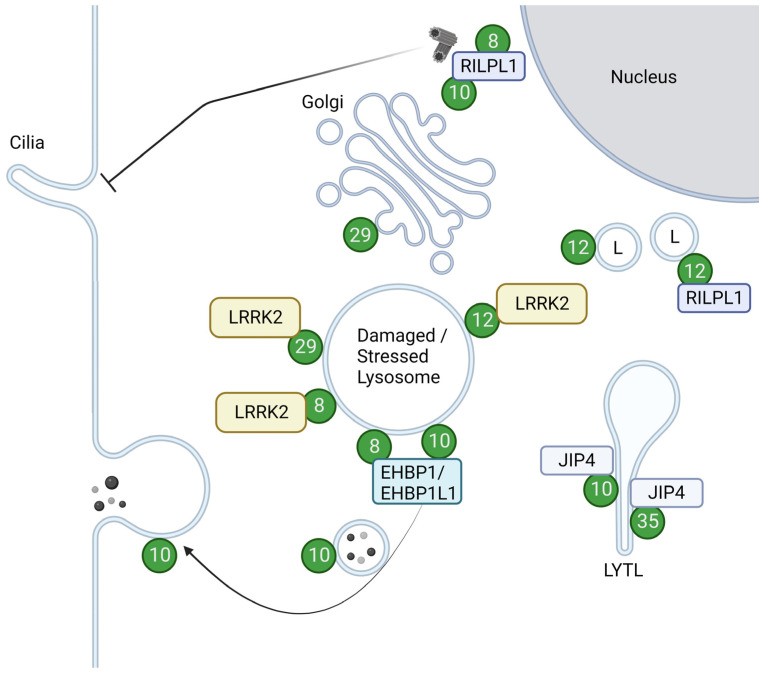
Localizations and functions of LRRK2-phosphorylated Rabs. Green circles with numbers indicate Rabs phosphorylated by LRRK2. LRRK2-phosphorylated Rab12 and Rab29 are recruited to damaged or stressed lysosomes and further activate LRRK2. LRRK2-phosphorylated Rab8 and Rab10, bound to their effector RILPL1, accumulate near centrosomes to inhibit ciliogenesis. Rab8 and Rab10 also act together with their effector EHBP1 or EHBP1L1 to counteract lysosomal inflation or facilitate lysosomal release. LRRK2-phosphorylated Rab10 and Rab35 bind to JIP4 and induce LYTL. LRRK2-phosphorylated Rab12 binds to RILPL1 and moves lysosomes to the perinuclear region. LRRK2-phosphorylated Rab29 alters the morphology of the *trans*-Golgi. L: Lysosomes. Figure created with BioRender.com.

**Table 1 biomolecules-13-01645-t001:** Studies on leucine-rich repeat kinase 2 (LRRK2) 3D structure.

Model of LRRK2	Method	Part of LRRK2 Analyzed	Interactions	Resolution	PDB ID	Year	Ref.
homodimer	Crystallography	LRRK2 ROC domain	Dimerization of ROC domain	2.0 Å	2ZEJ	2008	[46]
homodimer	Negative-stain EM	Full-length Strep/FLAG-LRRK2	-	22 Å	N/A	2016	[47]
homodimer	Cryo-EM	Full-length 3xFLAG-LRRK2	-	16 Å	N/A	2017	[48]
homodimer	Crystallography	LRRK2 WD40 domain	Dimerization of WD40 domain	2.6 Å	6DLO/ 6DLP	2019	[42]
filamentous	Cryo-ET	Full-length LRRK2 (I2020T)	COR:COR WD40:WD40	14 Å	6XR4	2020	[43]
filamentous	Cryo-EM	LRRK2 RCKW ^1^	COR:COR WD40:WD40	3.5 Å	6VNO (6VP6/6VP7 /6VP8)	2020	[40]
monomer homodimer	Cryo-EM	Full-length LRRK2	COR:COR	3.7 Å 3.5 Å	7LHW 7LHT	2021	[30]
monomer dimer tetramer (all with Rab29)	Cryo-EM	Full-length LRRK2	LRRK2:Rab29 COR:COR WD40:kinase	3.5 Å	N/A	2022	[44]
filamentous	Cryo-EM	LRRK2 RCKW ^1^	COR:COR WD40:WD40	5.0 Å	7THY/ 7THZ	2022	[49]

^1^ RCKW: LRRK2 C-terminal half containing ROC, COR, KIN, and WD40 domains.

**Table 2 biomolecules-13-01645-t002:** Summary of Rab GTPase phosphorylation by LRRK2.

Rab	Phosphorylation Site by LRRK2 ([68] Unless Otherwise Noted)	Effect of Phosphorylation
Rab1	Thr75 [69]	Unknown
Rab3	Thr86	Unknown
Rab5	Ser84, Thr6 (in vitro [70])	Delay in EGFR degradation [70]
Rab8	Thr72	Inhibition of ciliogenesis and centrosome cohesion [17,71,72] Decreased binding with Optineurin [73] Increased binding with RILPL2 [74] Mistrafficking of cargo to damaged lysosomes [75] Suppression of enlargement of stressed lysosomes [76] Activation of LRRK2 [52]
Rab10	Thr73	Inhibition of ciliogenesis and centrosome cohesion [15,71,77,78,79,80] Decreased binding to EHBP1L1 [81] Induction of LYTL by recruitment of JIP4 [82,83] Extracellular release of lysosomal contents under lysosomal stress [76] Lowers GCase activity [84] Induction of apoptosis after injury [85]
Rab12	Ser106	Perinuclear clustering of lysosomes via increase in RILPL1 binding [86] Activation and recruitment of LRRK2 [53,54]
Rab29	Thr71, Ser72	Regulation of trans-Golgi morphology [18] Recruitment of Rab29 itself to stressed lysosomes [87]
Rab35	Thr72	Induction of LYTL by recruitment of JIP4 [83] Increase in α-synuclein propagation [88]
Rab43	Thr82	Unknown

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
