# Peer review of "An Update on the Interplay between LRRK2, Rab GTPases and Parkinson’s Disease"

_biomolecules, 2023, doi:10.3390/biom13111645_

Round 1
Reviewer 1 Report
Comments and Suggestions for Authors
This is a useful compendium of recent findings related to LRRK2 activation by Rab GTPases. A few comments here are provided with the goal of improving the accuracy of the review.
1. Page 1. “to form specific areas” might be more clear to say “domains”
2. Page 2. LRRK2 seems inactive when not on membranes (it does not modify cytosolic Rab29 that is not GDI-bound) so it is unlikely that a cleavage product is functional; also, LRRK2 is highly sensitive to proteolysis and so the fragment may not be present in cells. Also, the affinity of the LRRK2 GTP binding site is low so that it will be occupied with the prevailing nucleotide (GTP) in the cytoplasm.
3. Page 4. Microscale thermophoresis was used in that work to monitor the binding of Rabs to the ARM domain and cannot provide information related to oligomeric state. The recruitment of LRRK2 to artificial planar lipid bilayers indicated some dimerization on the bilayer surface but should not be cited as the finding was not definitive. Also, Rab12 is now shown by two groups to provide significant activation of LRRK2 activity.
4. Page 4. The functions of Rabs are not really centered on the endolysosome system as many participate in the secretory pathway. Also, Rabs 8, 10 , 12 and 29 are not known as endolysosomal Rabs.
5. Table 2. There is no evidence to my knowledge that Rab5 is an actual LRRK2 substrate.
6. Page 10. Many reports show that G2019S LRRK2 increases Rab phosphorylation about two fold; the patient cells analyzed have high levels of phosphatases and proteases so detection of phosphorylation may be challenging.
7. A critical point that should be noted is that only a small fraction of Rabs is phosphorylated at steady state so that the majority of a given pool of Rab proteins can still carry out their normal functions (Ito et al., 2016).
Comments on the Quality of English LanguageIt would be useful for a native English speaker to help edit the text for smoother language usage.
Author Response
This is a useful compendium of recent findings related to LRRK2 activation by Rab GTPases. A few comments here are provided with the goal of improving the accuracy of the review.
>> Thank you very much for taking the time to review our manuscript and pointing out important issues.
- Page 1. “to form specific areas” might be more clear to say “domains”
>> We have changed the word here from “areas” to “domains”, following this suggestion.
- Page 2. LRRK2 seems inactive when not on membranes (it does not modify cytosolic Rab29 that is not GDI-bound) so it is unlikely that a cleavage product is functional; also, LRRK2 is highly sensitive to proteolysis and so the fragment may not be present in cells. Also, the affinity of the LRRK2 GTP binding site is low so that it will be occupied with the prevailing nucleotide (GTP) in the cytoplasm.
>> We agree with the view that this cleavage product lacking N-terminal membrane-interacting region is nonfunctional, so we have rewritten as such. We also agree that this product is unstable in cells, even based on our experience with cultured cells. On the other hand, the paper referenced here [Ref 38] reported that such product exists in macrophages as one of endogenous LRRK2 species and that it can heterodimerize with full-length LRRK2, so we considered that this product may not be negligible and have added such description.
- Page 4. Microscale thermophoresis was used in that work to monitor the binding of Rabs to the ARM domain and cannot provide information related to oligomeric state. The recruitment of LRRK2 to artificial planar lipid bilayers indicated some dimerization on the bilayer surface but should not be cited as the finding was not definitive. Also, Rab12 is now shown by two groups to provide significant activation of LRRK2 activity.
>> We realized that the statement “one of the active forms as heteromultimers” is confusing; this is followed by “with either multiple Rab8 and Rab10”, but the word heteromultimer evokes oligomer, so we have simply rewritten as “one of the active forms that bind either Rab8 or Rab10”.
As for two Rab12 papers, although we described them in a later section, we thought it would be better to introduce them here as well, right after the explanation of LRRK2-Rab29 complex, so we added the following sentence: “Of note, more recent studies have identified Rab12 rather than Rab29 as an activator of LRRK2, as described later, so a heteromultimer complex with Rab12 would also need to be assumed”.
- Page 4. The functions of Rabs are not really centered on the endolysosome system as many participate in the secretory pathway. Also, Rabs 8, 10, 12 and 29 are not known as endolysosomal Rabs.
>> We totally agree with this view and have rewritten as follows: “Their functions are associated with a wide variety of intracellular trafficking, ranging from cellular secretory pathways to intracellular degradation pathways involving the endolysosomal system.”.
- Table 2. There is no evidence to my knowledge that Rab5 is an actual LRRK2 substrate.
>> Phosphorylation of Rab5 at Ser84 has been described in two studies (Steger et al., Elife 2017 (Ref 15) and Lis et al., Biochem J 2018 (Ref 68)), although both studies showed phosphorylation under overexpression of Rab5 and did not succeed in detecting phosphorylation of endogenous species. Another study has merely reported the phosphorylation of Rab5b at Thr6 in vitro (Yun et al., J Biochem 2015 (Ref 70)), which seems more indeterminate. At this time, the authors Dr. Alessi and colleagues have not ruled out Rab5 as a substrate, so we have added the following sentence as a supplement to the Rab5 section (page 10): “Note that phosphorylation of Rab5 isoforms by LRRK2 has only been observed upon overexpression in cells [15,68] or in vitro [70], and therefore we need to be cautious as to whether they are indeed physiological substrates”.
- Page 10. Many reports show that G2019S LRRK2 increases Rab phosphorylation about two fold; the patient cells analyzed have high levels of phosphatases and proteases so detection of phosphorylation may be challenging.
>> Thank you for pointing this out. The difference in the effects of LRRK2 mutations is a very important point of discussion and should have been written more accurately. We have now modified the descriptions as follows (page 11, in the beginning of Section 4): “As stated above, almost all familial PD mutations increase Rab phosphorylation in cells, although the most common G2019S mutation may have somewhat different effects than others. That is, the G2019S mutation increases its intrinsic kinase activity more effectively than Rab phosphorylation, whereas the other mutations increase Rab phosphorylation more potently than its kinase activity [24]”.
- A critical point that should be noted is that only a small fraction of Rabs is phosphorylated at steady state so that the majority of a given pool of Rab proteins can still carry out their normal functions (Ito et al., 2016).
>> This is definitely another critical point, so we have added the following descriptions and discussions in the second paragraph of Section 5 (page 12): “As only a small fraction of Rabs is phosphorylated at steady state, it is likely that the majority of a given pool of Rab proteins can still carry out their normal functions [90]. On the other hand, phosphorylated Rab8a and Rab10 have been shown to bind new effector proteins, i.e., RILPL1/2 and JIP3/4 [15,74,83], suggesting that even a small fraction of them may dominantly affect cellular functions. Indeed, it has been shown that the recruitment of RILPL1 regulates ciliogenesis and centrosomal cohesion [15,71] and that of JIP4 regulates lysosomal tubulation [83] and axonal autophagosome transport [170]. In addition, as noted above, Rab10 phosphorylation by LRRK2 is markedly enhanced under lysosomal stress and is therefore assumed to play important roles in the maintenance of endolysosomes.”.
Comments on the Quality of English Language
It would be useful for a native English speaker to help edit the text for smoother language usage.
>> We have made some corrections to the text with the aid of Grammarly, an AI-based online English editing tool, throughout the manuscript.
Reviewer 2 Report
Comments and Suggestions for Authors
Komory and Kuwahara wrote a timely, concise and well-structured review on the link between LRRK2, Rab proteins and Parkinson's disease. I have some minor suggestions to improve it:
- Add references to the sentence: "endocytosis and intracellular transport involving the trans-Golgi network (TGN) and other organelles".
- It would be very helpful to add a figure representing the different locations/functions of the Rab proteins that are substrates for LRRK2.
- It would also be informative to expand the information on the roles of the Rab proteins in specific cell types that are relevant for Parkinson's disease, such as neurons and glia (for example PMID 34658337).
- Although the authors talk about kinase inhibitors in the last paragraph, given the clinical relevance of this topic, I suggest expanding this information to indicate how specific inhibitors affect Rab phosphorylation and in which cell types or models they have been tested.
Author Response
Komory and Kuwahara wrote a timely, concise and well-structured review on the link between LRRK2, Rab proteins and Parkinson's disease. I have some minor suggestions to improve it:
>> We appreciate this reviewer for reading our manuscript and raising valuable suggestions.
- Add references to the sentence: "endocytosis and intracellular transport involving the trans-Golgi network (TGN) and other organelles".
>> Thank you for pointing this out. This part was shared among the latter three phrases, which was not clear, so we have added the corresponding references to this phrase.
- It would be very helpful to add a figure representing the different locations/functions of the Rab proteins that are substrates for LRRK2.
>> We have added a new figure (Figure 2) and its legend related to Table 2, illustrating how each Rab protein functions upon phosphorylation by LRRK2 at various intracellular locations.
- It would also be informative to expand the information on the roles of the Rab proteins in specific cell types that are relevant for Parkinson's disease, such as neurons and glia (for example PMID 34658337).
>> We have added a whole paragraph explaining this point in Section 4, after 1st paragraph (page 11-12). Please look at the descriptions beginning with “In relevance to the pathomechanism of PD ...”.
- Although the authors talk about kinase inhibitors in the last paragraph, given the clinical relevance of this topic, I suggest expanding this information to indicate how specific inhibitors affect Rab phosphorylation and in which cell types or models they have been tested.
>> Thank you for this suggestion. We have now included the following sentences in the last paragraph of Section 5 (page 13): “According to their clinical trial reports, administration of this LRRK2 inhibitor has been shown to markedly reduce Rab10 phosphorylation in PBMCs, as well as phospho-Ser935 LRRK2 in whole blood, total LRRK2 in cerebrospinal fluid (CSF), and a lysosomal lipid di-22:6-bis (monoacylglycerol) phosphate (BMP) in urine, all in a dose-dependent manner [172]. Also, the clinical trial results of another LRRK2 inhibitor, DNL201, were reported earlier, and the results were mostly similar [107]”.